# Effectiveness of cognitive–behavioural therapy: a protocol for an overview of systematic reviews and meta-analyses

Beth Fordham,[1] Thavapriya Sugavanam,[1] Sally Hopewell,[2] Karla Hemming,[3] Jeremy Howick,[4] Shona Kirtley,[2] Roshan das Nair,[5] Julia Hamer-Hunt,[6] Sarah E Lamb[1]

For numbered affiliations see end of article.

**Correspondence to**
Dr Beth Fordham;
beth.fordham@ndorms.ox.ac.uk

## ABSTRACT

**Introduction** Cognitive–behavioural therapy (CBT) is a psychological therapy that has been used to improve patient well-being across multiple mental and physical health problems. Its effectiveness has been examined in thousands of randomised control trials that have been synthesised into hundreds of systematic reviews. The aim of this overview is to map, synthesise and assess the reliability of evidence generated from these systematic reviews of the effectiveness of CBT across all health conditions, patient groups and settings.

**Methods and analysis** We will run our search strategy, to identify systematic reviews of CBT, within the Database of Abstracts of Reviews of Effects, the Cochrane Library of Systematic Reviews, MEDLINE, Embase, PsycInfo, CINAHL, Child Development and Adolescent Studies, and OpenGrey between January 1992 and 25 April 2018. Independent reviewers will sift, perform data extraction in duplicate and assess the quality of the reviews using the Assessing the Methodological Quality of Systematic Reviews (V.2) tool. The outcomes of interest include: health-related quality of life, depression, anxiety, psychosis and physical/physiological outcomes prioritised in the individual reviews. The evidence will be mapped and synthesised where appropriate by health problem, patient subgroups, intervention type, context and outcome.

**Ethics and dissemination** Ethical approval is not required as this is an overview of published systematic reviews. We plan to publish results in peer-reviewed journals and present at international and national academic, clinical and patient conferences.

**Trial registration number** CRD42017078690.

## Strengths and limitations of this study

► A strength of this study is that it is the only up-to-date overview of systematic reviews examining randomised control trials of the effectiveness of cognitive–behavioural therapy (CBT) across all health problems, populations and settings.
► Another strength is that our method allows us to map the available evidence and identify where the evidence base is strong or weak.
► The main weakness is that we will only include systematic reviews that explicitly state, 'Cognitive behavioural therapy' (including all synonyms) in their abstract, title or keywords. This excludes broader reviews that encapsulate the CBT within 'psychological interventions'.
► Another weakness is that we are reliant on the information provided in the systematic reviews; therefore, we might omit randomised controlled trials if they are not included in the reviews we synthesised.

## INTRODUCTION

The cognitive behavioural model theorises that the way in which we think and behave will influence our emotional and physical well-being and consequently our overall quality of life. The relationships between cognitions, behaviours, emotions and physical responses are all considered bidirectional.[1,2]

Cognitive–behavioural therapy (CBT) is a talking therapy, which targets identifying maladaptive thoughts and behaviours and challenging them, trying to develop different ways of thinking and acting to improve the psychological and physical outcomes for patients. CBT has a core set of competencies that can be applied transdiagnostically; however, it has also been tailored for use in specific populations, such as CBT insomnia. Most CBT is delivered in adherence with CBT process manuals specific to the health problem. Roth and Pilling,[3] on behalf of the Department of Health, developed a set of core competencies for CBT and included a division between high and low intensity CBT. They defined high intensity as formal CBT with a CBT-trained health professional predominantly delivered face to face in an individual or group format. Low intensity interventions focus on patient self-help and can be delivered by health professionals with very little to fairly comprehensive CBT training and via several platforms (internet, phone and paper based). This distinction can become less clear in some forms of CBT, called 'blended care', where high-intensity therapy is combined with low intensity self-help methods.

The effectiveness of CBT has been evaluated with randomised control trials (RCTs), which have been synthesised into systematic reviews across numerous physical and mental health problems from schizophrenia[4] to low back pain.[5] We recognised some consistency across the CBT systematic reviews, for example, improving symptoms of insomnia in adults with various health problems.[6–8] However, we also identified areas with conflicting evidence for example with regards to the efficacy of CBT in reducing relapse in schizophrenia.[9 10]

While we are cognisant of the volume and variety of available systematic reviews of CBT, we are not aware of the quality of the reviews conducted across different health problems, populations and settings. Another limitation of current evidence is that short-term changes to function as a result of CBT do not guarantee long-term changes,[11 12] and much of the evidence focuses on shorter term outcomes.

This overview will explore the effects of CBT across all health problems, in all populations and in all settings. The primary outcome will be health-related quality of life (HRQL) with the aim of capturing, to some degree, the broader, general, biopsychosocial influence of CBT in addition to its impact on the specific functional outcomes.

## Rationale
Our scoping work suggests there are more than 500 systematic reviews of CBT, and there has been no published overview of systematic reviews since 2004. We aim to map for which populations there are systematic reviews of RCTs examining CBT and document how well these reviews were conducted. Within each population, we will identify whether: (A) there is a need for new or better quality systematic reviews or RCTs or (B) that CBT worsens/does not alter/improves generic (HRQL) and problem-specific health outcomes in comparison with active or not active control conditions in the short-term or long-term follow-up period.

## Objectives
The specific objectives include:
1. Stage one: a map of the evidence
   a. Map and assess the quality of available evidence.
2. Stage two: a synthesis of the evidence
   a. A descriptive and a panoramic meta-analytic (PMA) synthesis of the evidence by International Classification of Diseases-11 (ICD-11) health problem categorisations and by common outcomes (HRQL, depression, anxiety, psychosis and the most common physical/physiological outcome).
   b. Subgroup analysis to explore high versus low intensity CBT (as defined by Roth and Pilling[3]) for a health problem.

## METHODS
### Patient and public involvement
We are working with a CBT expert consultation group (ECG) consisting of clinical academics (n=7), research academics (n=9) and service users (n=4). We will meet with this group face to face twice and communicate via phone/email throughout the overview process to guide our protocol development, synthesis strategy and interpretation. We hope the ECG will guide our overview to produce clinically meaningful outputs. The group will not be involved in any of the data extraction or quality assessment to ensure no undue influence.

## Methods
We shall perform two stages within this overview. Stage 1 is to identify all the available systematic reviews of CBT, which include RCT evidence then to map the available evidence along with a quality assessment of the included reviews. The stage 2 will be to meaningfully synthesise the evidence by common outcomes across health problems and to specifically examine the comparative effectiveness of high and low intensity CBT.

### Stage 1: mapping the evidence
This stage will detail how we will identify and select the systematic reviews for inclusion in order to generate a comprehensive map of the evidence.

### Eligibility criteria
To be included in the evidence map and overview of systematic reviews, studies must meet the following criteria:

### Type of studies
We will include systematic reviews of RCTs that evaluate the effects of CBT. We will include systematic reviews that include both randomised and non-randomised trials so long as the review has summarised the RCT evidence independently.

To be included, systematic reviews must fulfil a minimum of four methodological criteria as defined by the Centre for Reviews and Dissemination (CRD), University of York, as part of the Database of Abstracts of Reviews of Effects (DARE) database (http://www.crd.york.ac.uk/crdweb)[13]:
1. inclusion/exclusion criteria reported,
2. adequate search strategy,
3. included studies synthesised,
4. quality of the included studies assessed,
5. sufficient details about the included studies reported.

The University of York has provided us with detailed definitions for each of these criteria. For example, the minimum sufficient details of the individual studies would be details of the population, setting, interventions and results for every included study (in text, tables or online appendices).

### Type of participants
We will include systematic reviews of RCTs, which include data from all age groups and any gender. We will include all health problems recognised within the ICD-11.

### Setting
We will include systematic reviews of RCTs that have been conducted in any context or country.

## Intervention

We will only include systematic reviews where CBT has been explicitly reported in the review title, abstract or keywords. We will include all formats of CBT. We will classify if the review's RCTs are employing high or low intensity CBT as defined by Roth and Pilling's Department of Health report.[3] High intensity CBT refers to face-to-face therapy with a relatively specialist trained CBT therapist and low intensity is all other types of CBT (blended care, guided self-help, internet-based, structured exercises or brief interventions).

## Comparator

We will include systematic reviews if they explore comparisons of CBT to either: (1) active: a non-CBT comparator intervention, placebo or treatment as usual; (2) no active: no intervention or waitlist control or (3) another format of CBT (eg, computerised CBT vs face to face).

## Outcomes

We will include systematic reviews that report information on at least one of the following patient or other reported outcomes

1. HRQL,
2. psychological,
3. physical/physiological.

   We will include reviews with short-term (<12 months) and long-term (≥12 months) outcomes.

## Restrictions

We will only include reviews that are published/available in the English language due to the limited study timescale. We shall only include reviews which were published after 1992.

## Information sources

Our method of identifying systematic reviews will be conducted according to the principles of the Cochrane Handbook for Systematic Reviews of Interventions[14] and recommendations for conducting Overviews of Systematic Reviews.[15]

The search strategy will be run across the DARE (up to March 2015), the Cochrane Library of Systematic Reviews, MEDLINE, Embase, PsycInfo, CINAHL, Child Development and Adolescent Studies, and OpenGrey. This list was compiled by testing and searching the specificity and inclusivity of several databases and with the guidance of the ECG.

## Search strategy

A comprehensive search strategy comprising free text and controlled vocabulary terms identified by the ECG and from key papers from our preliminary scoping searches of systematic reviews on CBT will be run. We will use The Scottish Intercollegiate Guidelines Network systematic review filter available on the InterTASC Information Specialists' Sub-Group website,[16] across MEDLINE, Embase and CINAHL. We will use the McMaster's filter[17] within PsycInfo.

Our scoping work has identified that the earliest published review of CBT, which has not been superseded, is 1992.[18] This year also saw the advent of the Cochrane Collaboration, which implements high-quality systematic reviews of RCTs across healthcare. Therefore, we restrict our search to the last 26 years.

Our search strategy picked up 36/36 sensitivity check papers. The strategy was adapted and checked for use across each of our selected databases. Our MEDLINE search strategy is attached in online supplementary appendix A.

We will perform an update search (April 2019) to check for any additional systematic reviews that have been published in the intervening year. We will also search PROSPERO, ClinicalTrials.gov and Clinical Trials Registry Platform (ICTRP) to identify any ongoing systematic reviews and clinical trials to inform our discussion.

## STUDY RECORDS

### Data management

Search results will be exported into Endnote for de-duplication and then exported into Covidence, as recommended by Cochrane.[19] The full text of reviews shortlisted for full-text analysis will also be uploaded to Covidence. We shall perform data extraction using Microsoft Excel.

### Selection process

Two reviewers will independently screen titles and abstracts using the abstract screening questionnaire, which is based on the eligibility criteria. We will obtain full-text reports of those reviews selected for inclusion or for any uncertain cases. Two reviewers will independently perform review selection with the full-text screening questionnaire, which includes the following reasons for exclusion:

1. not a systematic review,
2. does not summarise RCT data separately,
3. does not report CBT specific data separately,
4. CRD criteria (4 out of 5) not fulfilled,
5. no HRQL, psychological or physical/physiological outcome,
6. full text not available in English,
7. conference abstract with insufficient data.

We will not contact authors for clarification. We will resolve any disagreements regarding the inclusion or exclusion of individual reviews by discussion with a third reviewer.

The search process and study identification will be documented in a figure as recommended by Preferred Reporting Items for Systematic Reviews and Meta-Analyses statement.[20] This will result in a final list of included and excluded systematic reviews along with reasons for exclusion. This process will not be blinded so all reviewers will be able to see the authors and their affiliated institutions.

### Data collection process

We developed a bespoke data extraction form with the ECG. Two reviewers will pilot the form on the first 18 reviews from the sensitivity check for the search strategy

and revise accordingly. Two reviewers will extract the review data items and perform the AMSTAR-2 quality assessment. A third reviewer will compare the duplicate extractions, and the anomalies will be discussed until a decision is reached.

### Data items

The information extracted for each review will include study ICD-11 category of disease (primary or secondary level), aims, study design (systematic review of RCT or systematic review of RCT and non-RCT), risk of bias (note whether the review used a risk of bias measure), number of RCTs and number of participants, demographics, intervention and control group description (category (high or low intensity), number of RCTs and number of session/frequency, duration), setting and whether the review included HRQL, depression, anxiety, psychosis or a physical/physiological outcome (description). We shall make a free-text list of all available outcomes reported in the review, in addition to those we specifically target. Descriptive information on mechanism data, acceptability, satisfaction, adverse events and economic analyses will also be extracted, when available.

We shall therefore emphasise the importance of long term (≥12-month follow-up) above short term (<12-month follow-up).

### Critical appraisal of included reviews

Each systematic review will be assessed independently by two reviewers using the AMSTAR-2[21] tool. We will not reassess the quality of the individual included RCTs. We will calculate the rate of agreement between the two reviewers and report. We will resolve any discrepancies with a third reviewer. Guidance suggests there are seven critical domains within the AMSTAR-2 items[21] and suggests categorising a review with 'high' confidence in the results of the review if we find no critical weakness and no or only one non-critical weakness; 'moderate' confidence if more than one non-critical weakness with no critical weakness; 'low' if there is one critical weakness with or without non-critical weaknesses; and 'critically low' if there is more than one critical weakness with or without non-critical weaknesses.[21]

### Evidence map
#### Overall map

We will produce a Bubble map[22] to represent the volume of systematic review data across all physical and mental health problems. The map will denote the total number of reviews (size of bubble), the total number of participants included in the reviews (y-axis) and the number of RCTs (x-axis) by the primary physical or mental health (ICD-11 primary/secondary category) problem the review targets.

#### Mapping by health problem

Summary tables will present included review details grouped by ICD-11 categories. Information will include Intervention details, comparison group details, follow-up period, outcomes measured, effect size and CIs for primary outcome/outcome pertaining to aim of review, number of RCTs, AMSTAR-2 rating, age and country. Within each health problem category, we shall order reviews first by those that compared CBT to an active comparator and second those where it is compared with a non-active comparator.

#### Mapping by review details

The available evidence will also be described by the following:

1. severity (mild, moderate and severe),
2. who (children, adults and older adults),
3. how (CBT intervention details),
4. when (prevention, standard treatment, relapse prevention and so on),
5. where (primary, secondary and hospital setting),
6. anxiety
7. depression
8. psychosis
9. physical/physiological outcomes,
10. HRQL outcomes.

The table aims to show the areas where systematic reviews have looked and where they have not. We shall highlight any individual patient data meta-analyses. We propose to use the confidence ratings of AMSTAR-2[21] to code reviews with 'high confidence' (green), 'moderate confidence' (yellow), 'low confidence' (amber) and 'critically low' (red).[21] This aims to give some direction as to the level of confidence.

### Stage 2

From the evidence maps populated in stage 1, we shall focus on the common outcomes examined within the included reviews. Stage 2 is to identify systematic reviews that we can synthesise to identify generic and specific effects of CBT across and within health problems.

### Outcomes and prioritisation
#### Primary outcome

This overview will prioritise long-term effects of CBT on HRQL outcomes.

#### Secondary outcomes

Where no long-term (≥12 month) follow-up data are available, we shall present the longest follow-up point available or the time point where the meta-analytic synthesis was performed. If there are separate analyses for several measurements of the same outcome, then we will chose the analysis with the largest number of RCTs included. If they are equal, then we will select the analysis of the measurement with the best psychometric properties.

We shall always extract data on HRQL, depression, anxiety, psychosis and one physical/physiological outcomes. If, in addition to or instead of HRQL, depression, anxiety and psychosis, there are multiple psychological and physical/physiological outcomes, we will make a list of all available outcomes reported. If we find an additional common outcome, deemed meaningful by the

ECG, which we have not focused on, we can return to the review and extract this information.

If there are separate analyses for different classifications of response to treatment (response, recovery, relapse and remission) for the same outcome. We shall chose:

▶ That which is identified as the primary outcome.
▶ The analysis with the highest GRADE score (if available).
▶ The analysis that includes the greatest number of RCTs.
▶ Where available, we will descriptively report the descriptions of mechanisms of action, patient satisfaction, adverse events and economic outcomes.

## Selection process

We shall group all of the reviews that include an HRQL outcome together. From these, we shall identify those that have performed a meta-analysis of the data. These reviews shall be grouped by their ICD-11 categorisation (ie, neoplasms). At this stage, we shall check if any of the included systematic reviews, within a health problem category, share primary RCTs. If we identify two or more reviews, which are eligible for inclusion but share the same primary RCTs, we will use the following criteria hierarchy to choose one review for inclusion into the overview:
1. The review with the highest AMSTAR rating.
2. The most recent review.
3. The review with the larger number of studies included.

We shall return to the full text of reviews that are selected and extract effect sizes, CIs and heterogeneity measures. For effect sizes based on continuous outcome measures, the combined intervention/control group means, SD and the total number of participants per group shall be extracted. For binary outcomes, we shall extract from the combined intervention/control group the number of participants who have achieved the desired outcome plus the total number of participants.

The selected reviews will be examined to identify those with moderate clinical, design and statistical homogeneity. Statistical heterogeneity in treatment effect estimates between health problems will be explored using the $I^2$ statistic (moderate to low heterogeneity $I^2$ less than 75%); clinical heterogeneity will be explored through discussion with the ECG; and design heterogeneity will be explored using AMSTAR-2 scores.

We shall repeat this process for all reviews that include a depression outcome, an anxiety outcome and a psychosis outcome. We will list all the physical/physiological outcomes that have been examined across all of our included reviews. The outcome that is the most common will be identified as the fifth outcome for selection.

## Synthesis

We will synthesise these reviews and provide pooled treatment effects for all reviews that include a:
1. HRQL outcome,
2. Depression outcome,
3. anxiety outcome,
4. psychosis outcome,
5. most common physical/physiological outcome.

This formal quantitative data synthesis will be undertaken using a two-step frequentist approach to a PMA. This method provides a single pooled estimate of the treatment effect along with estimates of degree of heterogeneity between reviews. This allows for both between study variability within the health problem (if random effects meta-analysis was used in the original indication review) and between health problem variability (using random effects) but does assume exchangeability of treatment effects.

We will perform this process for the outcomes of HRQL, depression, anxiety, psychosis and the most common physical/physiological outcome. As we have collected other psychological and physical/physiological outcomes, we will remain flexible and will consider additional synthesis suggested by the ECG.

## Subgroup analysis

For each of our key outcomes (HRQL, depression, anxiety, psychosis and the most common physical outcome), we will perform a subgroup analysis comparing:
1. reviews that include RCTs with high intensity CBT,
2. those with low intensity CBT,
3. those with a mixture of high and low intensity CBT RCTs.

In addition, if we find reviews that directly compare high and low intensity CBT within the review, we shall group these and if possible pool the results, comparing high to low intensity CBT groups rather than intervention to control groups.

We do not plan to perform any further subgroup analyses; however, if the data are suitable, we are flexible to additional analyses (eg, by control group type or follow-up period) if the comparison is deemed important by the ECG once we have reviewed the available data.

## Publication bias

This will be assessed per outcome; therefore, if we have more than 10 systematic reviews per outcome (HRQL, depression, anxiety, psychosis and the most common physical outcome), then the evidence of funnel plot asymmetry will be assessed using both the funnel plot and the Egger test using a p value of 0.1 to acknowledge the low power of this test.

## Summary

We are sensitive to the importance of not overstating conclusions representing CBT as being effective or not and to accurately reflect where further research, whether primary or secondary analysis work, is needed. We will caveat all summary statements and recommendations with the limitations of the methodology but treat this as a necessary step in addressing the current state of the CBT evidence base.

The mapping exercise will identify in which health problems, across which subgroups, contexts and with what format, CBT has been evaluated, thereby identifying gaps that have not been examined with a high-quality systematic review.

The synthesis stage can identify if CBT can produce long-term changes in quality of life. It will also present, with varying degrees of confidence, where CBT does or does not produce generic or problem-specific long-term changes on specific functions.

We will search PROSPERO, ClinicalTrials.gov and ICTRP to identify ongoing, completed or published trials or systematic reviews that have addressed the areas we recommend for further research. This summary will lead to a set of recommendations regarding the prioritisation of primary or secondary research into areas where we cannot generalise the clinical effectiveness findings and the evidence base is weak.

### Dissemination plan

An overview of the project will be published in the National Institute for Health Research journals library. We plan to prepare secondary publications detailing the generic effects of CBT on HRQL, depression, anxiety, psychosis and the most commonly found physical/physiological outcome. When there is sufficient data, we will publish health problem-specific overview papers. We hope to present the findings at international conferences to make sure the information is communicated to the patient population perhaps via patient conferences and/or social media.

### Author affiliations
[1]Nuffield Department of Orthopaedics, Rheumatology and Musculoskeletal Sciences, University of Oxford, Oxford, UK
[2]Centre for Statistics in Medicine, Nuffield Department of Orthopaedics, Rheumatology and Musculoskeletal Sciences, University of Oxford, Oxford, UK
[3]Department of Public Health, University of Birmingham, Birmingham, UK
[4]Department of Primary Care Health Sciences, Centre for Evidence-Based Medicine, University of Oxford, Oxford, UK
[5]Institute of Mental Health, University of Nottingham, Nottingham, UK
[6]Department of Psychiatry, University of Oxford, Oxford, UK

**Acknowledgements** SEL receives funding from the National Institute for Health Research (NIHR) Collaboration for Leadership in Applied Health Research and Care Oxford at Oxford Health NHS Foundation Trust.

**Collaborators** Beth Fordham.

**Contributors** BF is the guarantor. TS developed the protocol with BF. SH provided expertise on design and methodology. KH provided expertise on methodology in particular panoramic meta-analysis. JH provided expertise in methodology. SK developed the draft search strategy plan. RdN and JH-H provided clinical contextual feedback. SEL provided expertise in all areas of design, methodology and rigour. All authors read, provided feedback and approved the final manuscript.

**Funding** This overview of systematic reviews is funded by the National Institute for Health Research, Health Technology Assessment (HTA) Programme (funding reference number HTA 15/174: Cognitive Behavioural Therapy: An overview of systematic reviews and meta-analyses). The University of Oxford is the sponsor of this overview of systematic reviews. The National Institute for Health Research (NIHR) HTA programme is funding this overview of systematic reviews. They will provide funding to Nuffield Department of Orthopaedics, Rheumatology & Musculoskeletal Sciences (University of Oxford) who will carry out the research and store the data. The NIHR HTA programme is not involved in any other aspect of the project design, data collection or analyses. They will be made aware of the findings at milestones throughout the project.

**Disclaimer** The views expressed are those of the author(s) and not necessarily those of the NHS, the NIHR or the Department of Health and Social Care.

**Competing interests** None declared.

**Patient consent** Not required.

**Provenance and peer review** Not commissioned; externally peer reviewed.

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
