## [Reviewer comments · BMJ Open]

ARTICLE DETAILS

TITLE (PROVISIONAL)	Effectiveness of Cognitive Behavioural Therapy: A protocol for an overview of systematic reviews and meta-analyses
AUTHORS	Fordham, Beth; Sugavanam, Thavapriya; Hopewell, Sally; Hemming, Karla; Howick, Jeremy; Kirtley, Shona; dasNair, Roshan; Hamer-Hunt, Julia; Lamb, Sarah

VERSION 1 – REVIEW

REVIEWER	Gerta Rücker Institute of Medical Biometry and Statistics, Faculty of Medicine and Medical Center - University of Freiburg, Germany
REVIEW RETURNED	02-Sep-2018

GENERAL COMMENTS	page 2, line 23: The acronym PS is unclear, probably the second author Thavapriya "Priya" Sugavanam is meant. Why not TS? p. 9, line 359: "a conservative P-value of 0.1": The p-value itself is okay (in fact it is recommended in tests of publication bias), but I would not call a P-value of 0.1 "conservative". Rather, it is a liberal value, as the probability of rejecting a true null hypothesis is high compared to the otherwise common 0.05. p. 9, line 376: "We will search Prospero [...] to identify on-going trials or systematic reviews": I suggest to replace "on-going trials" with "on-going, completed or published trials", as these trials also might have addressed some areas for further research.
--

REVIEWER	Daniel David Babes-Bolyai University/Romania and Icahn School of Medicine at Mount Sinai/New York
REVIEW RETURNED	04-Sep-2018

GENERAL COMMENTS	I like the protocol. There is need for some clarifications: (1) The first paragraph suggest that CBT is about how we react to a physical or mental disorder. This is not the case. CBT is also about how we develop mental disorders.(2) Psychosis outcome should be also targeted (both for theoretical and practical relevance).(3) Do you have hypotheses?;(4) Do you treat differently studies based on superiority vs. non-inferiority vs. equivalence logic?(5) Do you plan to treat differently the classical reviews/metaanalysis vs. IDP metaanalysis?
--

VERSION 1 – AUTHOR RESPONSE

Reviewer: 1

Reviewer Name: Gerta Rücker

Institution and Country: Institute of Medical Biometry and Statistics, Faculty of Medicine and Medical Center - University of Freiburg, Germany

Please state any competing interests or state 'None declared': None declared.

Please leave your comments for the authors below

page 2, line 23: The acronym PS is unclear, probably the second author Thavapriya "Priya" Sugavanam is meant. Why not TS? Thank you this was a mistake and has been amended.

p. 9, line 359: "a conservative P-value of 0.1": The p-value itself is okay (in fact it is recommended in tests of publication bias), but I would not call a P-value of 0.1 "conservative". Rather, it is a liberal value, as the probability of rejecting a true null hypothesis is high compared to the otherwise common 0.05. Amended – removed "conservative"

p. 9, line 376: "We will search Prospero [...] to identify on-going trials or systematic reviews": I suggest to replace "on-going trials" with "on-going, completed or published trials", as these trials also might have addressed some areas for further research. Amended – added "on-going, completed or published trials"

Reviewer: 2

Reviewer Name: Daniel David

Institution and Country: Babes-Bolyai University/Romania and Icahn School of Medicine at Mount Sinai/New York

Please state any competing interests or state 'None declared': CBT supervisor

Please leave your comments for the authors below

I like the protocol. There is need for some clarifications:

(1) The first paragraph suggest that CBT is about how we react to a physical or mental disorder. This is not the case. CBT is also about how we develop mental disorders. Thank you I have amended the sentence to make it clearer, "The cognitive behavioural model theorises that the way in which we think and behave will influence our emotional and physical well-being and consequently our overall quality of life."

(2) Psychosis outcome should be also targeted (both for theoretical and practical relevance). Thank you we agree, we have informed the HTA of our plan to include psychosis as an outcome and have amended the protocol throughout.

(3) Do you have hypotheses?; The HTA did not request us to specifically state a priori hypotheses as our aim was to primary map and where possible synthesise the existing evidence base.

(4) Do you treat differently studies based on superiority vs. non-inferiority vs. equivalence logic? We are using the reviews rather than the RCTs as our entry data therefore it depends on how a review has treated these individual RCTs. We will simply report what is available from the review. We have not planned for any further analysis.

(5) Do you plan to treat differently the classical reviews/metaanalysis vs. IDP metaanalysis? If we encounter IDP meta-analyses we shall identify them as such in the mapping exercise however it is outside of our scope to conduct any further analysis, "We shall highlight any Individual Patient Data (IDP) meta-analyses."

VERSION 2 – REVIEW

REVIEWER	Gerta Rücker Institute of Medical Biometry and Statistics, Faculty of Medicine and Medical Center - University of Freiburg, Germany
REVIEW RETURNED	15-Oct-2018
GENERAL COMMENTS	In my view the authors have responded appropriately to all reviewers' comments.